# Structural insights into AQP3 channel closure upon pH and redox changes reveal an autoregulatory molecular mechanism

Peng Huang [1,6], Raminta Venskutonytė [1,2,6], Carter J. Wilson [3,6], Sara Bsharat[4], Rashmi B. Prasad [4], Pontus Gourdon [1,5], Isabella Artner [4], Bert L. de Groot [3] & Karin Lindkvist-Petersson [1,2] ✉

Regulation of intracellular levels of reactive oxygen species (ROS) remains poorly understood. Aquaporin 3 (AQP3) facilitates the membrane transport of hydrogen peroxide ($H_2O_2$), a key ROS signaling molecule. Here we elucidate the molecular mechanism of AQP3 and show that its regulatory properties are both pH dependent and autoregulated by $H_2O_2$. Using single particle cryo-electron microscopy, we present open and closed conformations of human AQP3. At pH 8.0, the channel adopts an open state, while acidic pH or exposure to $H_2O_2$ promotes closure via a large conformational rearrangement of extracellular loop E. These findings reveal a mechanism for autoregulation of $H_2O_2$ transport and establish AQP3 as a key modulator of redox homeostasis in human pancreatic β-cells.

Reactive oxygen species (ROS), including superoxide and hydrogen peroxide ($H_2O_2$), are generated as byproducts of cellular metabolism. In the absence of appropriate antioxidant mechanisms, cells become overwhelmed by oxidative stress, resulting in cellular dysfunction and even cell death[1]. Tight regulation of intracellular ROS is therefore essential, and disruptions in redox homeostasis are implicated in numerous human diseases. Human pancreatic β-cells are particularly vulnerable to oxidative stress due to their relatively low expression of antioxidant enzymes, including catalase, superoxide dismutase, and glutathione peroxidase[2,3]. This limited antioxidant capacity may reflect a physiological requirement for $H_2O_2$ as signalling molecule in glucose stimulated insulin secretion (GSIS)[4]. For example, NADPH oxidase 4 (NOX4)- generated $H_2O_2$ has been directly linked to GSIS in β-cells[4]. NOX generated $H_2O_2$ requires membrane channels, aquaporins (e.g., AQP3), to rapidly cross cell membranes[5]. Of the 13 human aquaporins, only the genes for *AQP3* and *AQP7* are highly expressed in human islets based on RNA-seq data from 188 human donors (Islet Gene View)[6,7]. Unlike AQP7, AQP3 facilitates the transport of $H_2O_2$ and is gated by pH, fully active at neutral pH and completely inactive at pH 6 or lower[8].

Notably, intracellular acidification has been observed in artificial β-cells under hyperglycaemic conditions[9] in addition to the more well-known acidification of the interstitial fluid in diabetic patients[10], suggesting that the activity of AQP3 may change upon glycemic alterations and possibly effect the transmembrane flux of NOX generated $H_2O_2$ in human pancreatic β-cells. In addition, glucose in chronic excess causes intracellular oxidative stress, possibly connected to AQP3 activity, which in turn results in defective insulin gene expression and insulin secretion as well as increased apoptosis in pancreatic β-cells[11].

Here, we report single-particle cryo-EM structures of human AQP3 under high (pH 8.0) and low (pH 5.5) pH conditions, and in the presence of $H_2O_2$. AQP3 adopts the canonical aquaporin fold with glycerol molecules occupying the channel pore at pH 8.0. By contrast, under acidic conditions, AQP3 assumes a fully closed conformation, preventing glycerol entry. Remarkably, exposure to $H_2O_2$ at pH 8.0 induces a similar closed state, indicating that $H_2O_2$ can promote channel closure independently of pH. Both closed structures show a dramatic collapse of extracellular loop E into the pore, effectively

[1]Department of Experimental Medical Science, Lund University, Lund, Sweden. [2]LINXS - Institute of Advanced Neutron and X-ray Science, Lund, Sweden. [3]Computational Biomolecular Dynamics Group, Max Planck Institute for Multidisciplinary Sciences, Gottingen, Germany. [4]Lund University Diabetes Centre, Clinical Research Center, Malmo, Sweden. [5]Department of Biomedical Sciences, Copenhagen University, Copenhagen N, Denmark. [6]These authors contributed equally: Peng Huang, Raminta Venskutonytė, Carter J. Wilson. ✉e-mail: karin.lindkvist@med.lu.se

sealing the channel and preventing solute passage. Combined with molecular dynamics simulations, the structures identify key residues involved in the pH-sensing mechanism and reveal how $H_2O_2$ stabilizes the closed state. Together, these findings provide a structural framework for pH-dependent gating of AQP3 and suggest an autoregulatory mechanism by which AQP3 modulates $H_2O_2$ flux across membranes.

## Results

### The structure of open and closed human AQP3

To elucidate the molecular mechanism of AQP3, the structure of AQP3 was resolved by single particle cryo-EM at an overall resolution of 3.3 Å (Fig. 1a). The protein was reconstituted into nanodiscs and imaged at pH 8.0. Cryo-EM data processing resulted in 2D classes with clear views of a tetrameric protein (Figs. 1a and S1). A 3D reconstruction using C1 symmetry preserved potential asymmetries between monomers and yielded a high-quality map suitable for model building (Figs. 1a and S1). Each of the four chains presented well defined cryo-EM densities for model reconstruction allowing complete modeling of each chain except for the flexible termini (Fig. 1a). Each AQP3 monomer showed the canonical aquaporin fold of a right-handed bundle consisting of six transmembrane helices (TM1-TM6) and two half-membrane spanning helices (Fig. 1a). Five loops are connecting the helices, with loop A, C, E on the extracellular side, and loop B and D on the intracellular side

(Fig. 1b). Cryo-EM densities consistent with glycerol were found in all four channels (Figs. 1a and S2) and glycerol molecules were modeled at these sites. For most molecules, the real-space correlation coefficients were 0.7–0.8 (Phenix), supporting their assignment. Although the exact conformations and positions of the glycerols within the channel could not be resolved, we have modelled the molecules based on the expected hydrogen bond interactions and previous knowledge of aquaporin-glycerol structures. To facilitate glycerol permeation, the characteristic aromatic/arginine (ar/R) selectivity filter (formed by Phe63, Tyr212, and Arg218) is widened compared to orthodox aquaporins, with a 7 Å separation between the arginine and tyrosine residues. The structure aligns closely with previously reported structures of human AQP7[7,12] and the recent cryo-EM models of rat AQP3[13]. Well-defined cryo-EM density near the conserved NPA motifs (formed by Asn215 and Asn83) is observed in all four channels, consistent with previous structures of AQP7 and AQP10, where glycerol molecules are bound at this site (Figs. 1a and S2)[7,12,14]. The free energy profile for glycerol permeation through AQP3 reveals a pronounced minimum near the NPA motifs and closely aligns with the positions observed in the cryo-EM density (Fig. 1c). The ar/R filter represents the highest energetic barrier for both water and glycerol permeation, with barriers of ~10 kJ/mol and ~23 kJ/mol, respectively. Assuming a Kramers-style rate, the barriers suggest that glycerol permeates roughly 500 times

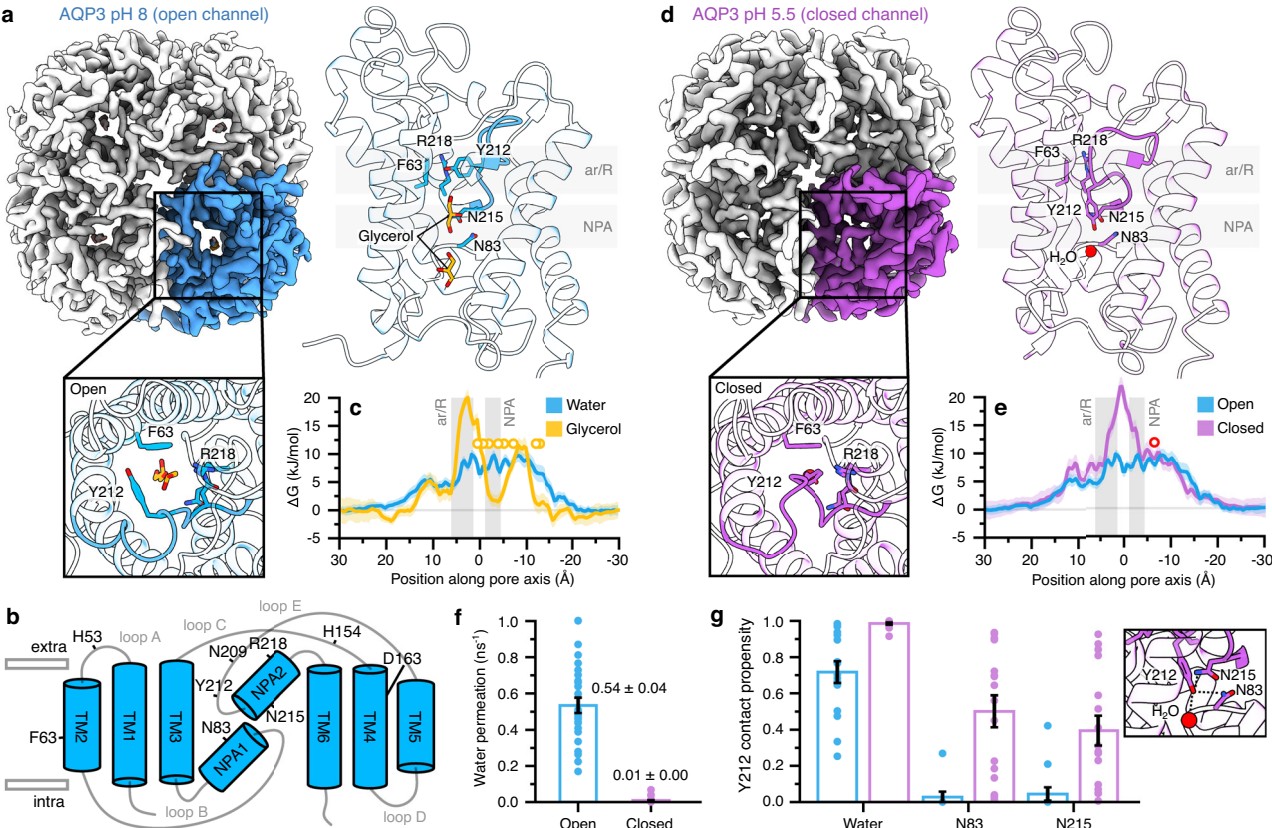

**Fig. 1 | The open and closed state of AQP3. a** Left: Cryo-EM density of AQP3 at pH 8.0 (view from extracellular side), with one monomer highlighted in blue, right: cartoon side view of a single AQP3 monomer at pH 8.0, showing modeled glycerol molecules (yellow) and key residues of the ar/R selectivity filter and NPA motifs (cyan), as sticks. Inset shows cartoon views down the pore axis. **b** Schematic representation of AQP3 topology with transmembrane helices and loops labeled. Residues forming the ar/R filter and NPA motifs are indicated. **c** Free energy of solute permeation through the pore of the open structure; SEM shown as bootstrapped error band over the 4 monomers. Yellow circles correspond to cryo-EM resolved glycerol z-positions over all four monomers. **d** Left: Cryo-EM density of AQP3 at pH 5.5 (view from the extracellular side), with one monomer highlighted in

magenta, right: cartoon side view of a monomer at pH 5.5, showing modeled water molecules (red) and ar/R and NPA motif residues as sticks (magenta). Inset shows cartoon views down the pore axis. **e** Free energy of water permeation through the open and closed structures; SEM shown as bootstrapped error band over the 4 monomers. Red circle corresponds to cryo-EM resolved water z-position in the closed structure. **f** Water permeation rates from MD simulations of open and closed AQP3. Rates calculated for each monomer and replicate are shown as points; SEM was estimated via bootstrap. **g** Interaction propensity of the Tyr212 hydroxyl group with NPA motif residues and an intra-channel water molecule. Propensities calculated for each monomer and replicate are shown as points; SEM was estimated via bootstrap over 4 replicates and 4 monomers.

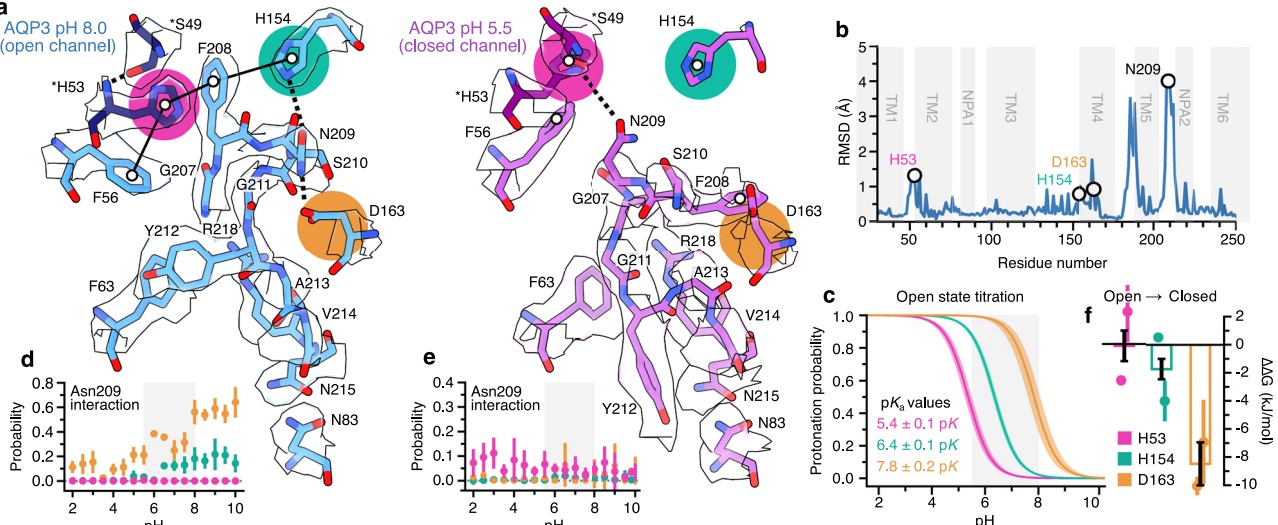

**Fig. 2 | Loop E collapses into the channel and seals the pore. a** Stick representation of loop E and adjacent residues in the open (blue) and closed (magenta) conformations. His53 and Ser49 are marked in a darker shade to represent that they belong to the neighboring monomer. The key titratable interaction partners of Asn209 are highlighted. The hydrophobic tetrad in the open structure is marked with a black line, and hydrogen bonds are marked as dashed lines, and supporting cryo-EM density in white. **b** Residue-wise root mean squared deviation (RMSD) of backbone atoms between the open and closed structures. Transmembrane helices and NPA motifs (Asn83, Asn215) are labeled; Asn209, which undergoes the largest conformational shift, is indicated. **c** CpHMD titration curves for His53, His154, and Asp163 in the open conformation. Curves correspond to Henderson-Hasselbach titrations implied by pKa values averaged over 4 monomers and 6 replicates; SEM shown as bootstrap error bands. **d** pH-dependent interaction propensities between Asn209 and His53, His154, and Asp163 in the open state across the CpHMD simulations. **e** Same as (**d**), but for the closed state. **f** Relative free energy change between open and closed conformations upon protonation of each titratable residue. Values from CpHMD and alchemical calculations are shown as individual points with corresponding errors; bar heights and SEM were computed from these values, with errors propagated.

slower than water, in good agreement with previous experimental measurements[13]. Together, these findings highlight the prolonged residence times for glycerol in the NPA region—a hallmark feature of aquaglyceroporins[7,12]. The tetrameric assembly of AQP3 forms a central pore (Fig. 1a), where we identified an extended cryo-EM density that was modeled as a fos-choline-8 molecule, which was present in the cryo-EM sample (Fig. S2). Densities in the central pore have been observed in AQP5, AQP7, AQPZ, and plant aquaporins[7,15–17]. Although the physiological function of the central pore remains unresolved, the presence of ligands in multiple aquaporin structures raises the possibility of a biological role.

AQP3 activity is reversibly inhibited by an acidic pH, in the range of pH 5.5 to 7.0[8,18,19]. To investigate the structural basis for pH sensitivity, nanodisc-reconstituted AQP3 was adjusted to pH 5.5 prior to cryo-EM grid freezing. Cryo-EM data processing resulted in 2D classes representing a tetrameric protein, similar to the pH 8.0 assembly (Figs. 1d and S3). The final cryo-EM map was reconstructed at 3.5 Å with C1 symmetry, as well as at 3.2 Å with C4 symmetry. As no variation in ligand or protein conformation was observed among the individual monomers in the C1 model, further analysis was only carried out with C4 symmetry, which yielded a higher-quality map (Figs. 1d and S3) and enabled confident modeling of the backbone and non-terminal side chains (Fig. 1d). Strikingly, the channel adopted a closed conformation, marked by a large-scale rearrangement of extracellular loop E (residues 207–212) that fully blocks the pore (Fig. 1d). In the closed conformation, Tyr212, part of the ar/R selectivity filter, shifts into the pore, positioning its hydroxyl group near the NPA motifs (Fig. 1d). A similar closed conformation of rat AQP3 with Tyr212 blocking the channel pore was recently reported (Fig. S4)[13]. HOLE analysis suggests that the collapsed loop creates a complete block, with all predicted pathways exhibiting near-zero pore radii (Fig. S5). Consistent with this physical obstruction, the free energy profile of water permeation through the pore indicates a barrier of 23 kJ/mol at the Tyr212 constriction region, as well as a local minimum that corresponds to the water density

observed in the cryo-EM structure (Fig. 1e). Unbiased MD simulations further confirm functional closure, showing robust water permeation in the open state ($0.54 \pm 0.04$ ns$^{-1}$ per monomer) and negligible permeation in the closed state ($0.01 \pm 0.00$ ns$^{-1}$ per monomer) (Fig. 1f). In the closed state, water molecules accumulated asymmetrically around Tyr212, the latter being stabilized by hydrogen bonds with the NPA motifs and an intra-channel water molecule (Fig. 1g), which agrees with the cryo-EM data.

## Disruption of tetrad stacking and Asp163 protonation favor channel closure

Analysis of structural rearrangements upon loop E closure revealed key interactions governing the gating. In the open (pH 8.0) structure, Phe208 forms a stacking interaction with His154 and lies in proximity to His53 from the neighboring monomer (Fig. 2a). His53 is stabilized by a hydrogen bond to Ser49 and stacks with Phe56, completing a HFHF tetrad (His154, Phe208, His53 and Phe56) that stabilizes the extracellular loop (Fig. 2a). Upon closure (pH 5.5), this interaction network is completely lost. Phe208 undergoes a dramatic shift, flipping from the intermonomer space into the channel (Fig. 2a), while the cryo-EM density of the loop containing His154 becomes poorly defined, consistent with instability following disruption of the HFHF tetrad (Fig. S6a). Zelenina and co-workers previously reported that His154Ala and His53Ala substitutions resulted in an impermeable channel or decreased permeability, whereas the His154Phe substitution had no significant effect, suggesting a role for aromatic stacking in stabilizing AQP3[8]. Our open structure reveals that these residues participate in a HFHF tetrad, providing a mechanistic explanation for their functional importance. Notably, this stacking interaction involves His53 from a neighboring monomer, underscoring the importance of the conserved AQP tetrameric organization (Fig. S6a).

Beyond disruption of tetrad stacking, His53 and His154 are titratable residues positioned to contribute directly to AQP3's pH-gating mechanism (Fig. 2b). In the open (pH 8.0) structure, His154 can form

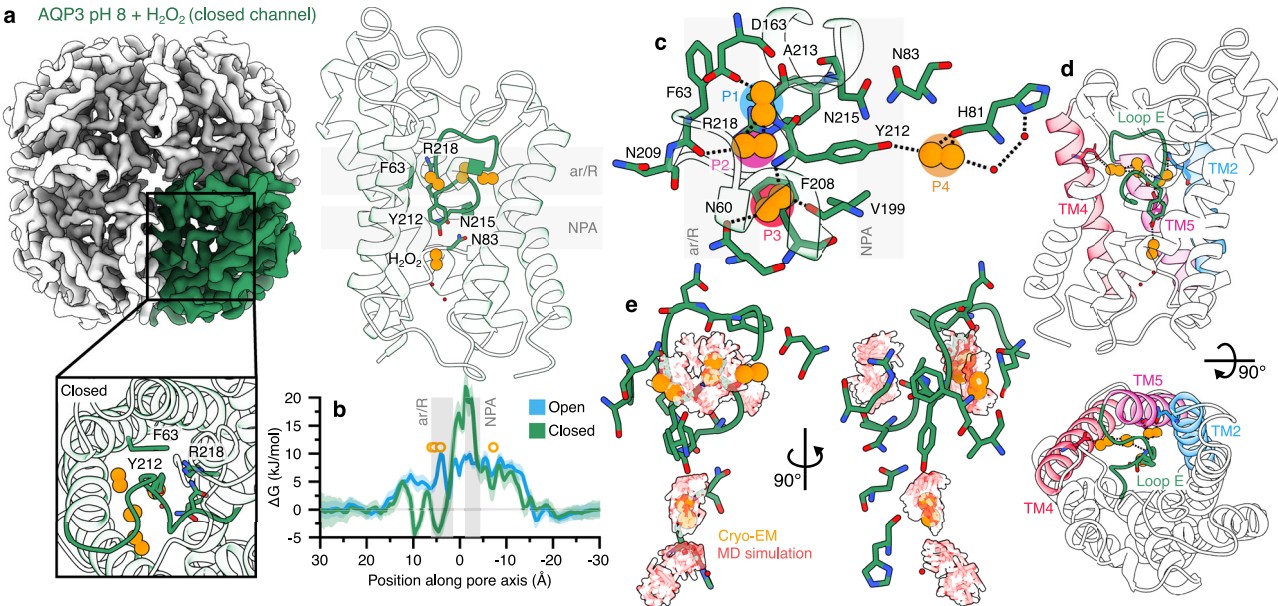

**Fig. 3 | AQP3 in the presence of H₂O₂. a** Left: Cryo-EM density of AQP3 at pH 8.0 (view from extracellular side), with one monomer highlighted in green. Right: Cartoon side view of a single AQP3 monomer at pH 8.0, showing modeled H₂O₂ molecules (orange) and water (red) and key residues of the ar/R selectivity filter and NPA motifs as sticks. Inset shows cartoon views down the pore axis. **b** Free energy of H₂O₂ permeation through the open and closed pore; SEM shown as bootstrapped error band over the 4 monomers. **c** Four modelled H₂O₂ molecules in the cryo-EM structure and their hydrogen bonding network with the collapsed loop E are indicated as dashed lines. **d** Position of the H₂O₂ interaction network relative to the transmembrane helices (red, purple, and blue) and loop E (green), upper: side-view and lower: extracellular side. **e** Overlay of cryo-EM modelled H₂O₂ molecules (orange) and H₂O₂ clusters identified from unbiased molecular dynamics simulations (white).

hydrogen bonds to Asn209 (directly adjacent to Phe208), while in the closed (pH 5.5) structure Asn209 shifts to form a hydrogen bond with His53 (Fig. 2a). Additional titratable residues near loop E, including Asp163 and Asp219, may also contribute since in the open conformation Asp163 forms a hydrogen bond with Asn209, which is lost upon channel closure. Constant-pH molecular dynamics (CpHMD) simulations indicate that the apparent p$K_a$ of Asp219 is downshifted in both open and closed conformations, suggesting it is unlikely to contribute to pH gating under physiological conditions. In contrast, His53, His154, and Asp163 exhibit apparent pKa values ranging from 5.4 to 7.8 pK, consistent with roles in pH-sensitive regulation (Fig. 2c). Protonation-dependent contact analysis revealed that interactions between Asn209 and both Asp163 and His154 are sensitive to pH, with contact propensities decreasing markedly at lower pH (Fig. 2d). Notably, the Asn209–His154 interaction occurred with appreciable frequency only when Asn209 was also engaged with Asp163, indicating a cooperative hydrogen-bonding network. Protonation of Asp163 disrupts its contact with Asn209, indirectly destabilizing the Asn209–His154 interaction. In the closed conformation, these pH-sensitive interactions were diminished, though Asn209–His53 sidechain contacts increased slightly with decreasing pH, consistent with the structural data (Fig. 2e). Hence, the simulations further support the functional importance of the HFHF tetrad, revealing increased stacking interactions between His53–Phe208 and Phe208–His154 with increasing pH in the open state and disruption in the closed conformation (Fig. S6b).

To directly quantify how pH influences the conformational equilibrium of AQP3, we computed the deprotonation free energies of His53, His154, and Asp163 in both open and closed states. Based on CpHMD titration curves and independent free energy calculations, average monomer destabilization energies upon deprotonation were −0.12 ± 1.09, −1.72 ± 0.73, and −8.46 ± 1.53 kJ/mol for His53, His154, and Asp163, respectively (Fig. 2f). These values indicate that the protonation of Asp163 favors the closed state by approximately 8.5 kJ/mol. The underlying basis for this destabilization is that Asp163 is buried, with a low solvent-accessible surface area (Fig. S7), and lacks nearby charge-

charge stabilizing interactions, instead forming a hydrogen bond with Asn209 (Fig. 2a). This environment raises the free energy of deprotonation, shifting the apparent pKa in the open state to 7.8 ± 0.2, substantially elevated relative to the intrinsic aspartate pKa of 3.9 and placing it within the physiological pH range (Fig. 2d). In the closed state, solvent accessibility increases slightly (Fig. S7), but the loss of the stabilizing Asn209 interaction further increases the deprotonation free energy. Together, the cryo-EM structures and simulations support a model in which channel closure is initiated by protonation of Asp163. This disrupts its interaction with Asn209, destabilizing the Asn209–His154 contact and prompting reorientation of loop E residues. Phe208 and Asn209 subsequently switch positions, breaking the inter-monomer HFHF tetrad and initiating a cascade of movements involving Ser210, Gly211, and ultimately Tyr212 (part of the ar/R selectivity filter), which collapses into the channel and forms a seal that occludes permeation (Fig. 2a).

## AQP3 channel is closed in the presence of hydrogen peroxide

In addition to transporting water and glycerol, AQP3 efficiently facilitates the flux of hydrogen peroxide (H₂O₂). To investigate how AQP3 responds to H₂O₂, we determined its structure in the presence of 150 μM H₂O₂ (1:1 protein/H₂O₂ molar ratio) at pH 8.0. AQP3 reconstituted in nanodiscs yielded well-defined 2D classes of the tetramer, and ab-initio reconstruction followed by refinement produced a 3.0 Å map in C4 symmetry (Figs. 3a and S8). The resolution enabled confident modeling of the backbone and non-terminal side chains (Figs. 3a and S8). Strikingly, AQP3 adopted a conformation identical to the acidic (closed) structure (RMSD of 0.23, calculated in Pymol), with extracellular loop E collapsing into the channel and occluding the pore (Fig. 3a). In the open channel, the height of the maximum barrier for H₂O₂ permeation occurs at the ar/R filter and was roughly 10 kJ/mol (Fig. 3b). In the closed channel, collapse of loop E shifts and increases the permeation barrier to -28 kJ/mol.

Analysis of the cryo-EM map revealed four cryo-EM densities in the vicinity of the upper part of loop E and within the closed channel pore near Tyr212 (Fig. S9). Interestingly, such densities were not

present in the closed structure map in the absence of $H_2O_2$, supporting the presence of $H_2O_2$ molecules in these positions. Three $H_2O_2$ molecules (P1–P3) were modeled near the collapsed loop E, forming a hydrogen-bonding network both among themselves and with surrounding residues, including those in the transmembrane helices (Fig. 3a, c, d). Specifically, P1 forms hydrogen bonds with Asp163 in TM4 and with P2. P2, in turn, connects to the carbonyl oxygen of Phe208 in loop E and to the backbone nitrogen of Ala213. P3 hydrogen bonds to the backbone nitrogen of Tyr212 as well as to the side-chain of Asn60 in TM2 and to the carbonyl oxygen of Val199 in TM5. Collectively, these interactions link loop E to the transmembrane core as P2 stabilizes residues in loop E forming an interaction bridge to TM4 via P1, while P3 directly connects loop E to TM2 and TM5 (Fig. 3c, d). The fourth $H_2O_2$ molecule (P4) was assigned to the cryo-EM density beneath the Tyr212 (Fig. S9). It forms a hydrogen bond with Tyr212 and extends the interaction network toward His81 via two bridging water molecules (Fig. 3d). While the cryo-EM densities and correlation coefficients for each $H_2O_2$ molecule (0.78, 0.65, 0.76 and 0.84 for the P1, P2, P3 and P4, respectively) support their presence at the assigned positions, we also performed MD simulations to further validate the $H_2O_2$ binding sites. Unbiased molecular dynamics simulations of the open and closed conformations in the presence of $H_2O_2$ revealed several stable clusters of bound $H_2O_2$, including two dominant clusters near the collapsed loop E and one beneath Tyr212 (Fig. 3e). Indeed, the positions of these clusters are in excellent agreement with the cryo-EM densities assigned to $H_2O_2$ (Fig. 3e). Furthermore, the free energy profile of permeation indicates that collapse of loop E creates a favorable binding region for $H_2O_2$, with an affinity of −5 kJ/mol relative to bulk (Fig. 3b). The absence of this binding region in the open state supports a role for $H_2O_2$ in stabilizing the closed conformation. Our simulations do not suggest a well-defined $H_2O_2$ binding site; rather, they suggest a binding region near the collapsed loop E, which likely involves multiple peroxide and water molecules. Binding of $H_2O_2$ within this region, in turn, stabilizes the protein's closed conformation. Together, these results indicate that $H_2O_2$ can promote channel closure even at neutral pH and suggest a mechanism of autoregulation in which excess $H_2O_2$ binding stabilizes the closed conformation of AQP3 by anchoring loop E. This ligand-induced gating provides a structural basis for feedback regulation of $H_2O_2$ transport, positioning AQP3 as both a facilitator and sensor of redox signaling in cells.

### AQP3 expression in human pancreatic islets from type 2 diabetic donors

To evaluate the potential role of AQP3 in human metabolism, AQP3 expression was analyzed in RNAseq data from liver, fat, islet, and muscle samples ($n = 12$). AQP3 expression was detected in all tissues, with the highest expression in liver and islet samples (Fig. 4a). Given the low catalase activity in islets compared to the liver[20], and the involvement of NOX4-derived $H_2O_2$ in glucose-stimulated insulin secretion (GSIS) in pancreatic β-cells[4], we focused subsequent analyses on AQP3 in human islets. A comparison of AQP3 mRNA levels from normoglycemic and type 2 diabetic donor islets in a larger cohort ($n = 188$) showed that AQP3 expression was elevated in type 2 diabetic islets (Fig. 4b, $p$ value = 0.16; the lack of significance is likely due to the small sample size of T2D donors). In contrast, catalase expression levels, which are already low in β-cells from normoglycemic donors, were further decreased in type 2 diabetic donors (Fig. 4c, $p$ value = 0.012). To determine whether the increase of AQP3 expression is reflected at the protein level, immunohistochemistry on pancreatic sections was performed from both donor groups using antibodies against AQP3 and islet hormones. AQP3 protein was co-expressed with insulin and glucagon and appeared more abundant in β-cells from type 2 diabetic donors (Fig. 4d). Upon quantification of AQP3 expression levels in islets from normoglycemic ($n = 3$) and T2D ($n = 4$) donors, a slight non-significant increase in AQP3 protein

expression was observed ($p = 0.18$) (Fig. 4e). Quantification of immunofluorescence revealed a large variation of signal intensity between different type 2 diabetic donors ($p < 0.0001$). Unfortunately, we could not associate the variation in AQP3 expression to a specific trait due to the low number of type 2 diabetic donors available and large variation in donor characteristics (like age, BMI, disease onset, etc).

To further evaluate if genetic variation at the *AQP3* gene locus is associated with islet expression, type 2 diabetes, or metabolic traits associated with blood glucose homeostasis (i.e., fasting blood glucose), we conducted genome wide association data look ups and analyzed islet specific expressions using RNA-seq data from human islets ($n = 420$) provided by the ISPIRE consortium. These analyses identified a genome-wide eQTL signal (rs60223038) and several nominal signals for *AQP3* (Data File 1, indicated in red)[21]. Notably, the genome-wide SNP is also a β-cell eQTL[21], and rs60223038 is nominally associated with fasting blood glucose ($p$ value = 0.000994). Several additional SNPs were identified as nominal islet eQTLs and were also nominally associated with fasting blood glucose levels (Data File 1, indicated in blue). Interestingly, a rare missense variant in *AQP3* (Arg223Cys; minor allele frequency is 0.00003) was identified and found to be nominally associated with fasting glucose ($p$ value = 0.008). Arg223 is located within the extracellular vestibule of the channel, where it forms a salt bridge with Asp219, a residue situated between Arg218 and Arg223 of the ar/R selectivity filter (Fig. 4f). Substitution of Arg223 with cysteine is predicted to disrupt this salt bridge, altering Asp219's interaction network and potentially destabilizing the ar/R filter and gating mechanism. Taken together, these findings suggest that AQP3 contributes to the regulation of blood glucose levels and that rare missense mutations may impair its regulatory properties, supporting a central overall role for AQP3 in islet function.

## Discussion

AQP3 is well established to facilitate the transport of glycerol, $H_2O_2$, and water at neutral pH, while becoming inactive at acidic pH[5,8,18]. The loss of permeability at acidic pH has been shown to be rapid and reversible and is well established by functional assays[8,18], suggesting the presence of a gating mechanism. Here, we used single particle cryo-EM in combination with molecular dynamics simulations to define the molecular basis for AQP3 gating. Our findings show that at low pH, protonation of Asp163 initiates a cascade of structural rearrangements that drive channel closure. Specifically, protonation of Asp163 disrupts its hydrogen bond to Asn209, causing a shift in Asn209 that in turn disrupts the HFHF tetrad (His154, Phe208, His53, Phe56) that stabilizes the open conformation. The critical role of this tetrad is supported by functional data as substitution of His53 or His154 with alanine renders AQP3 impermeable, while His154Phe substitution (still capable of maintaining stacking interactions) preserves partial function[8]. Similarly, substitution of Ser49 to alanine, which supports the tetrad via hydrogen bonding, abolishes channel permeability[8]. Notably, the HFHF tetrad is structurally conserved in human AQP7, but in contrast to AQP3, AQP7 remains active at both neutral and acidic pH[18]. A key difference is that AQP7 has an asparagine in the position of Asp163, supporting the conclusion that protonation of Asp163 triggers channel closure in AQP3. Our data show that at physiological pH, Asp163 exhibits about 70% protonation, increasing to roughly 90% just below neutral pH (Fig. 2c). This suggests that AQP3 exists in a dynamic equilibrium between open and closed states in vivo and may close in response to subtle decreases in pH, which is supported by the recent structure of rat AQP3 that adopts a closed conformation at pH 7.5[13]. Unexpectedly, human AQP3 also adopted a closed conformation in the presence of 150 μM $H_2O_2$ at pH 8.0 (Fig. 3), indicating that elevated $H_2O_2$ levels can stabilize the closed state. Together, these findings suggest that under physiological conditions, AQP3 dynamically samples between the open and closed conformations, with both mild

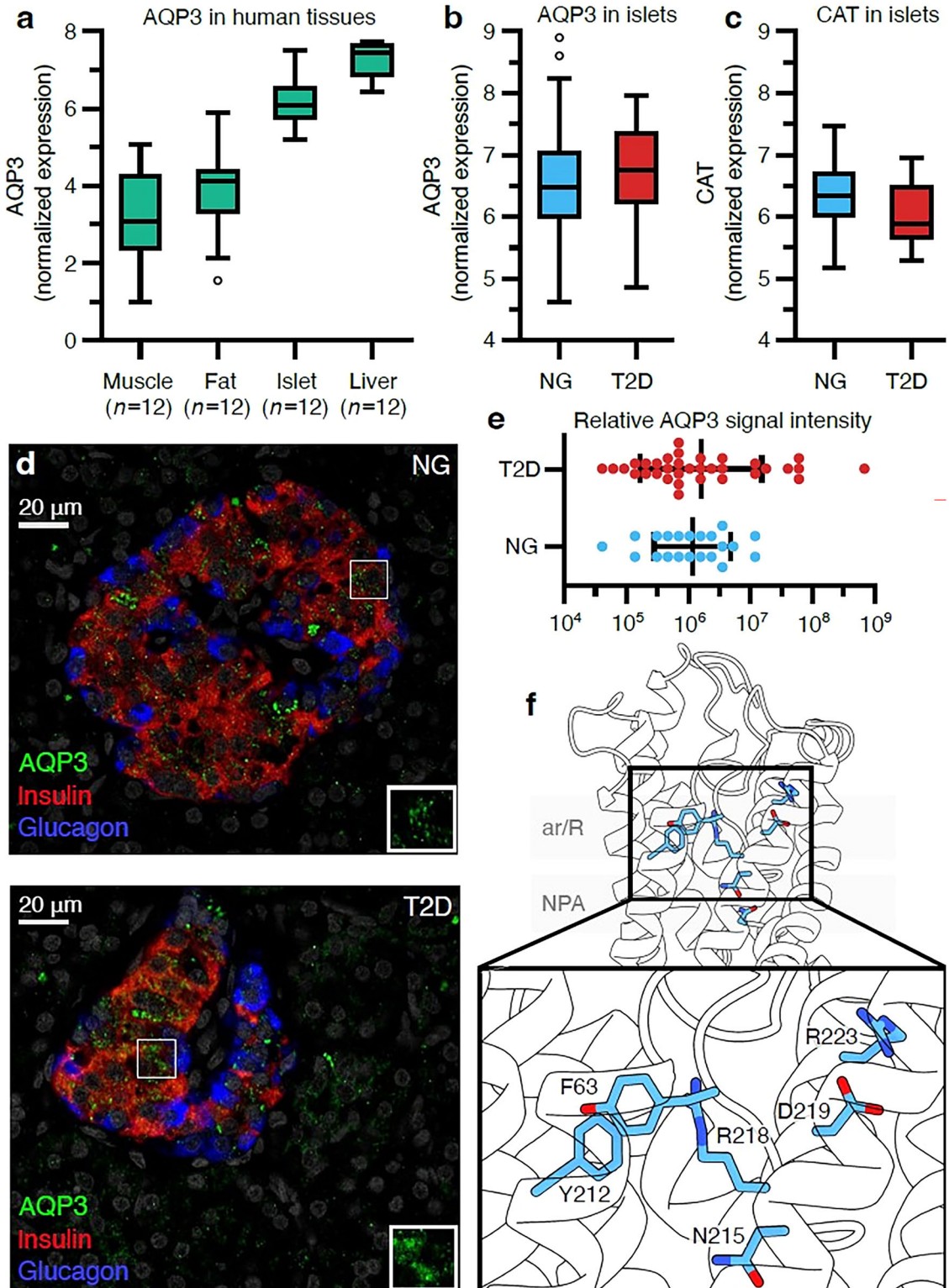

**Fig. 4 | AQP3 expression in human pancreatic islets. a** AQP3 expression in human tissues. **b** AQP3 expression in islets from normoglycemic donors (blue) and donors diagnosed with T2D (red). **c** Catalase expression in islets from normoglycemic donors (blue) and donors diagnosed with T2D (red). $n = 188$ for (**a**) to (**c**). **d** Immunohistochemistry staining for (**a**) AQP3 (green), insulin (red), and glucagon (blue), in human pancreas sections from (upper) normoglycemic donors and (lower) donors diagnosed with T2D, scale bar 20 μM. **e** Quantification of AQP3 expression levels in islets from normoglycemic (blue, $n = 3$) and T2D (red, $n = 4$) donors. **f** Close-up view of the ar/R filter in AQP3 with Arg223, Asp219, and Arg218 shown as sticks (cyan). Boxplots extend from Q1 to the Q3 with the mean depicted as a separate line. The whiskers extend 1.5 times the interquartile range.

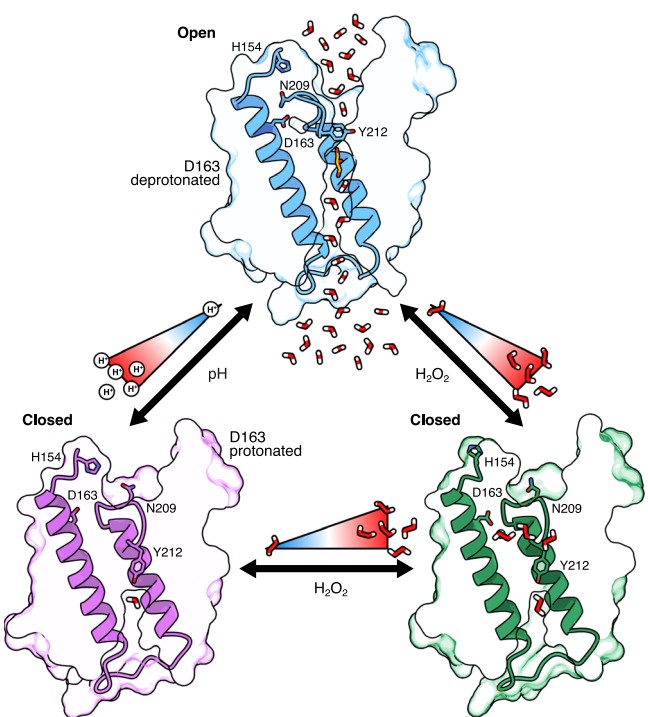

**Fig. 5 | Proposed AQP3 gating mechanism.** The equilibrium between the open state (blue), closed state (magenta), and a closed state with bound $H_2O_2$ (green) is governed by the changes in pH and the presence of $H_2O_2$.

acidification and increased $H_2O_2$ concentrations shifting the equilibrium towards channel closure (Fig. 5).

Indeed, previous studies have shown that AQP3 facilitates $H_2O_2$ uptake in a concentration-dependent manner in mammalian cells[5,22]. However, at higher extracellular concentrations (in excess of 100 μM), this effect is lost, $H_2O_2$ uptake becoming indistinguishable between AQP3-expressing cells and non-transfected controls[5,22]. These findings suggest that while low micromolar concentrations of $H_2O_2$ are efficiently transported via AQP3, elevated concentrations may trigger autoregulatory channel closure, stabilizing the inactive conformation. Such a mechanism would be broadly protective, limiting excessive ROS flux across membranes. In most tissues, intracellular $H_2O_2$ levels are kept low by antioxidant enzymes such as catalase, which decomposes $H_2O_2$ into water and oxygen[23]. However, in pancreatic islets, the situation is different, as catalase gene expression is relatively low[21] and $H_2O_2$, produced by NOX-family enzymes[24], can function as a signalling molecule to promote glucose-stimulated insulin secretion (GSIS) in β-cells[4]. In type 2 diabetes, chronic hyperglycemia elevates ROS, which contributes to β-cell dysfunction. The slightly elevated levels of AQP3 mRNA and protein observed in type 2 diabetic islets may reflect a compensatory attempt to reduce intracellular $H_2O_2$ levels. However, as pH levels decrease during hyperglycemia[10], protonation-induced closure of AQP3 further limits $H_2O_2$ efflux, exacerbating oxidative stress and contributing to the dysfunction of type 2 diabetic β-cells. This model is further supported by the presence of risk alleles at the AQP3 locus, which are associated with altered islet gene expression and impaired glycemic control. Interestingly, catalase expression, already low in β-cells from normoglycemic donors, is further reduced in type 2 diabetic donor islets, likely exacerbating intracellular ROS accumulation and contributing to β-cell dysfunction[6]. Together, our structural findings reveal a protonation-driven gating mechanism for AQP3 and position it as a redox-sensitive regulator of $H_2O_2$ flux. This, in combination with genetic associations linking AQP3 to blood glucose control, supports a regulatory role for AQP3 function in physiological and pathological pancreatic β-cells.

## Methods

Human tissue was obtained from the Human Tissue Laboratory, which is funded by the Excellence of Diabetes Research in Sweden (EXODIAB) network (www.exodiab.se/home) in collaboration with the Nordic Network for Clinical Islet Transplantation Program (www.nordicislets. org). Written informed consent was obtained from pancreatic donors or their relatives, and all procedures were approved by the Swedish Ethical Review Authority (Permit number 2011263).

### AQP3 expression and purification
The double-deletion strain of *P. pastoris* (with both endogenous aquaporins deleted) containing the full-length human aquaporin 3 (AQP3) gene fused with C-terminal poly-histidine was used to express AQP3[18]. Yeast cells expressing AQP3 constructs were cultured, and membranes were prepared according to published protocols with minor modifications[25]. The isolated membrane fraction was washed with 20 mM NaOH and solubilized in the buffer containing 20 mM Tris-HCl, pH 8, 500 mM NaCl, 1% DDM, 10% glycerol, 1 mM DTT, and protein inhibitor cocktail (Thermo Fisher Scientific) for 1 h at 18 °C. Insoluble material was spun down by ultracentrifugation, the supernatant collected and incubated with Ni-NTA resin (Invitrogen) for 16 h before loading onto a gravity chromatography column (Bio-Rad). The resin was washed with 20 mM Tris-HCl pH 8, 500 mM NaCl, 20% glycerol, 30 mM imidazole, and 0.05% DDM to remove non-specifically bound protein, and the AQP3 eluted in the same buffer but including 500 mM imidazole. The eluted protein was concentrated and further purified by size exclusion chromatography in a buffer consisting of 20 mM Tris-HCl pH 8, 150 mM NaCl, and 0.05% DDM (Superdex 200 Increase 10/ 300 GL, Cytiva). Finally, eluted fractions containing the tetrameric AQP3 were pooled together and concentrated for nanodisc reconstitution.

### MSP1E3D1 production and purification
The plasmid encoding MSP1E3D1 was transformed into BL21(DE3) cells, followed by protein expression and purification according to published procedures[26,27]. Briefly, the cells were lysed by sonication in the PBS buffer supplemented with DNase I (Roche) and protein inhibitor cocktail (Thermo Fisher Scientific). Thereafter, the lysate was cleared by centrifugation, and the supernatant was further incubated with Ni-NTA resin (Invitrogen) for 16 h. The protein was purified by washing the resin consecutively with buffer I (40 mM Tris-HCl pH 8, 300 mM NaCl, 1% Triton 100), buffer II (40 mM Tris-HCl pH 8, 300 mM NaCl, 50 mM Na-cholate, 20 mM imidazole) and buffer III (40 mM Tris-HCl pH 8, 300 mM NaCl, 50 mM imidazole), and further eluting in buffer consisting of 40 mM Tris-HCl pH 8, 300 mM NaCl and 400 mM imidazole. The eluted protein was incubated with Tobacco Etch Virus (TEV) protease and dialyzed in Tris-buffer (20 mM Tris-HCl, pH 8, 100 mM NaCl) for 16 h to cleave the His tag before once more incubating with Ni-NTA to remove the His-tagged TEV protein.

### Nanodisc reconstitution
AQP3 protein was reconstituted into nanodiscs comprised of the scaffold protein MSP1E3D1 and POPC lipids at a molar ratio of 1:4:77 (AQP3:MSP1E3D1:POPC). Briefly, AQP3 protein in DDM was incubated with POPC at 18 °C for 30 min, supplemented with MSP1E3D1 and 100 mg Bio-Beads resin (Bio-Rad), and incubated for 1 h. Later, the mixture was replaced with new Bio-Beads, followed by the incubation of 1 h to remove DDM completely. Afterwards, the solution was loaded onto a Ni-NTA resin packed column (Invitrogen) to remove empty nanodiscs once the Bio-beads were discarded. The Ni-NTA resin was washed with 20 mM Tris-HCl, pH 8, 100 mM NaCl, and nanodisc-reconstituted AQP3 was eluted in the buffer containing 20 mM Tris-HCl, pH 8, 100 mM NaCl, and 300 mM imidazole. The eluted protein was concentrated and subjected to size exclusion chromatography (Superdex 200 Increase 10/300 GL, Cytiva) in 20 mM Tris-HCl buffer at

pH 8 supplemented with 100 mM NaCl (Fig. S10). Finally, the AQP3-containing peak fractions were pooled and concentrated to 2 mg/ml for cryo-EM grid preparation. To create an acidic environment, the AQP3 protein (7 mg/ml) prepared as described above, was pH-adjusted to 5.5 using 1 M MES buffer (pH 5.5) prior to freezing on the cryo-EM grid. For the AQP3 sample containing hydrogen peroxide, $H_2O_2$ was added to a final concentration of 150 μM to AQP3 (pH 8, 4.7 mg/ml) immediately before grid preparation.

## Cryo-EM grid preparation and data collection
Three samples were plunge-frozen onto cryo-EM grid in presence of 0.5 mM fluorinated Fos-Choline-8 (Anatrace) at pH 8.0. The same grid type (Quantifoil R 2/1 300 mesh, gold) glow-discharged by 20 mA, 60 s, was applied for all three samples with 3 s blotting and 3 s waiting using a Vitrobot Mark IV (FEI) at 4 °C with 100 % humidity. The data collection was performed on the 300 kV Titan Krios electron microscope (FEI) equipped with a Ceta-D camera and a Gatan K3 BioQuantum detector and energy filter.

In the data collection for AQP3 sample at pH 8.0, the total dose was 39.866 e⁻/Å² fractionated across 40 frames with exposure time of 1.7 s at the pixel size of 0.85 Å. Finally, 4179 movies were acquired at the magnification of 105000x with the nominal defocus range from −0.8 to −2 μm by using software EPU (v2.12.1). The data collection for AQP3 at acidic pH and in the presence of $H_2O_2$ was collected using the same parameters with the total dose 52.825 e⁻/Å² over 40 dose fractions in the exposure time of 1.8 s, leading to a pixel size of 0.83 Å. Finally, 11820 and 15001 micrographs were collected for two grids, respectively, using the same nominal defocus range and magnification as AQP3 at pH 8.0.

## Cryo-EM data processing and model building
All datasets were processing using cryoSPARC (version 4)[28]. The collected micrographs were subjected to patch motion correction and patch CTF estimation, and poor images were excluded by running curate exposure before further processing.

For AQP3 pH 8.0 dataset, the template was initially created by 2D classifying the particles picked manually, which was further applied to all 3286 micrographs for particle picking by template-based auto-picking tool. Particles were extracted by using box size of 300 pixel and averaged by running several cycles of 2D classification. Subsequently, good quality particles (approximately 233 583) were selected from 2D classes, and further ab-initio reconstructed and classified into 3 classes at the 3D level. The most likely 3D class reconstructed using 139,031 particles representing 60% of all particles was further refined by non-uniform refinement and CTF refinement, resulting in two maps with nominal resolution of 2.9 Å and 3.3 Å at FSC value of 0.143 when applying C4 and C1 symmetry, respectively. The local resolution was run and evaluated in cryoSPARC.

For AQP3 low pH dataset, 8075 micrographs with optimal parameters were selected by running curate exposure for further process. The initial template created by blob-picking job was applied to pick particles across 8075 micrographs, resulting in almost eight millions selected particles. Subsequently, all these particles were aligned and screened by running several cycles of 2D classification, and the best particles were selected for three-dimensional ab-initio reconstruction and classification. Later, heterogeneous refinement was performed for three classes generated from ab-initio reconstruction, from which the best class with clear helical features was further refined by non-uniform refinement, generating two maps with nominal resolution at 3.2 Å (C4 symmetry) and 3.5 Å (C1 symmetry), respectively.

The data process for $H_2O_2$ dataset was performed by applying initial templates created in previous AQP3 data processing onto all selected micrographs by using template picker tool. Furthermore, all extracted particles were classified by running 4 rounds of 2D classification, and selected good classes were reconstructed and refined on the 3D level. Finally, the potential map for AQP3 in this dataset was refined to a resolution at 3.0 Å by applying either C4 or C1 symmetry.

The maps were used to fit the model structures in Chimera[29], and afterwards the models were inspected and edited in Coot[30] and refined using real-space refinement in Phenix[31]. Data collection, processing, and refinement parameters and statistics are provided in Table S1. The cryo-EM maps for all the structures allowed to model most of the amino acids, except for the termini. For the pH 8.0 structure, cryo-EM densities were assigned to glycerol and FF8 detergent, and in the closed AQP3-$H_2O_2$ structure, four $H_2O_2$ molecules were modelled. In the AQP3-$H_2O_2$ structure we have also observed extra densities within the pore, where water molecules were modelled, forming the interaction network with $H_2O_2$ and Tyr212. When assigning ligand positions, the decisions have been made by evaluating cryo-EM densities by taking into account the surrounding well-defined protein regions and also calculating the real-space correlation coefficients for each ligand (Table S2). In all the structures, additional densities surrounding the transmembrane part of the tetramer could be seen, which most likely represent lipid molecules used in nanodisc reconstitution of the protein.

## In silico analysis
The HOLE software[32] (http://www.holeprogram.org/) was used for pore dimension analysis. Calculations for all AQP3 structures were performed with an end radius of 5 Å. Visualization was performed in PyMOL (Schrödinger) and Chimera[29]. Plotting was done using matplotlib[33], and figures were organized in Inkscape.

## Molecular dynamics simulations
All simulations were performed using plain GROMACS[34] or a patched constant-pH version[35,36] with the CHARMM36m force field[37] and CHARMM-modified TIP3P water[38], which has been previously used to simulate aquaglyceroporins[39]. Individual system configurations were prepared using Charmm-GUI[40], embedding the open and closed AQP3 tetramers into lipid bilayer patches of 168 POPC lipids (additional details in Table S3). Default GROMACS protonation states were used. Sufficient potassium and chloride atoms were added to neutralize the system and achieve a 150 mM ionic strength. Energy minimization was performed using the steepest descents algorithm. Temperature was maintained at 300 K using velocity rescaling[41] ($\tau = 1.0$ ps), while the pressure was maintained at 1 bar using the C-rescale barostat[42] ($\tau = 5.0$ ps). A 2.0 fs was used. Long-range electrostatic interactions were calculated using Particle-mesh Ewald (PME)[43] (Fourier spacing: 0.12 nm, real-space cut-off of 1.0 nm). Lennard-Jones interactions were force-switched off between 1 nm and 1.2 nm. Bond lengths were constrained using the Parallel LINear Constraint Solver[44].

Rate-based permeabilities were calculated assuming a Kramers-style form of the rate and using the potential of mean force barrier height. Specifically,

$$k \approx \frac{D}{\delta^2} e^{-\frac{\Delta G^{\ddagger}}{k_B T}}$$

where $D$ is the in-pore diffusion assumed to be 3 times slower for glycerol than water[45], is pore length, i.e., 5 nm, $G^{\ddagger}$ is the maximum barrier height, and $k_B T$ is thermal energy.

For constant-pH molecular dynamics, a multisite representation for titratable residues was employed for aspartate, glutamate, and histidine. The mass of the λ particle was set at 5 AU, and its temperature was maintained at 300 K using velocity rescaling with a 2 ps coupling time. The barrier height of the double-well potential was set at 5.0 kJ/mol. 100 buffer particles were used. Six production simulations of between 0.5 and 1.5 μs were performed. The first 100 ns were discarded as equilibration.

For the alchemical free energy calculations, generation of initial structures and topologies was performed using pmx[46]. Following previous protocols[47–49], a double system in a single box setup was used, with a 3 nm distance between the AQP3 tetramer and capped peptide (ACE-AXA-NH$_2$) to ensure charge neutrality during the alchemical transition. A single Cα in the peptide was positionally restrained to prevent interaction with the tetramer. Simulations used a leap-frog stochastic dynamics integrator[50,51] ($T = 300$ K, $\gamma = 0.5$ ps$^{-1}$) and were run for 300 ns in 4 independent replicas, and the first 50 ns of each simulation was discarded as equilibration. From the remaining 250 ns, 250 non-equilibrium transitions of 500 ps each were generated, and work values from the forward and backward transitions were collected using thermodynamic integration. We observed a good overlap in the work distributions (Fig. S11). Bennett's acceptance ratio[52] as a maximum likelihood estimator relying on the Crooks fluctuation theorem[53] was used to estimate the relative free energies of deprotonation, and uncertainties were estimated via bootstrapping.

For the umbrella sampling simulations, initial configurations were generated by either pulling the solute along the channel axis or manually shifting its z-position. Umbrella windows were spaced at 1 Å, with a 2000 kJ/mol/nm$^2$ harmonic restraint applied to the solute's z-coordinate. Lateral restraints were imposed using flat-bottom potentials in the $x$ and $y$ directions: zero within a 10 Å radius from the channel axis and harmonic ($k = 1000$ kJ/mol/nm$^2$) beyond. Each window was simulated for 100 ns, discarding the first 10 ns as equilibration. The solute in each monomer channel was treated as an independent replicate. Potential of mean force (PMF) profiles were computed from the resulting probability distributions using the weighted histogram analysis method (WHAM).

### AQP3 expression analysis in human pancreatic islets
Gene expression from 188 donors was processed as described previously[6]. Human pancreas was obtained from the Human Tissue Laboratory at the Lund University Diabetes Center, which is funded by the Excellence of Diabetes Research in Sweden (EXODIAB) network (www.exodiab.se/home) in collaboration with the Nordic Network for Clinical Islet Transplantation Program (www.nordicislets.org). Informed consent was obtained from pancreatic donors or their relatives, and all procedures were approved by the Swedish Ethical Review Authority (Permit number 2011263). Post-processing of expression data was performed by alignment to reference genome build 37, and gene counts were computed using feature counts. Counts were further normalized for sequencing depth, and rank-based inverse normal transformation was applied. Linear models were applied to assess the association between gene expression and genotypes with age, sex, BMI, purity, and days in culture as covariates. eQTLs and GWAS lookups of type 2 diabetes and related traits (fasting blood glucose levels) at the *AQP3* locus were performed in the AMPT2D portal (Type 2 Diabetes Knowledge Portal - Home (hugeamp.org)) and INSPIRE dataset[21]. Briefly, RNAseq and GWAS data from human pancreatic islets from 420 donors, and FAC-sorted β-cells from 11 donors[54] were processed as described previously[21]. Briefly, reads were mapped to the GRCh37 reference genome with GEM and genes, and exons were quantified from GENCODE annotation v19. Imputed genotypes were related to gene expression using fastQTL with a window of 1 Mb for cis-eQTLs, using the first 4 PCs for genotypes, the first 25 PCs for expression, gender, and a variable identifying the laboratory of origin of the samples.

### Immunohistochemistry
Immunohistochemical analysis was performed as described previously[55]. Human pancreas biopsies from normoglycemic and type 2 diabetic donors were fixed in 4 % paraformaldehyde and embedded in paraffin, and 6-μm sections were mounted on glass slides. The primary antibodies with the following dilutions were used: insulin (1:1000, Dako, Cat#506442), AQP3 (1:100, Abcam, Cat#ab125219), and glucagon (1:1000, Abcam, Cat#ab10988). Secondary antibodies used were: Cy2-, Cy3-, and Cy5- conjugated α-guinea pig, α-mouse, and α-rabbit (1:500. Jackson ImmunoResearch, Cat#s: 711-165-152, 712-176-153, 706-225-148, 715-225-150, 715-175-150). Nuclear staining was performed using DAPI (1:6000, Invitrogen). For quantification, formalin-fixed paraffin-embedded pancreatic sections from three normoglycemic and four type 2 diabetic donors were stained with the following primary antibodies: anti-insulin (guinea pig, 1:1000), anti-glucagon (mouse, 1:2000), and anti-AQP3 (rabbit, 1:100). Islets were identified based on insulin and glucagon staining. Images were acquired at 60× magnification using identical microscope settings (Nikon AX) for all samples. The mean fluorescence intensity of AQP3 within the entire islet area was measured using ImageJ/Fiji. The islet region of interest (ROI) was manually delineated based on the insulin and glucagon channels, and the corresponding mean AQP3 signal intensity was recorded. Background intensity was measured from an adjacent unstained tissue area and subtracted from the islet AQP3 intensity. The relative AQP3 intensity was compared between normoglycemic and type 2 diabetic donors using Welch's *t*-test.

### Reporting summary
Further information on research design is available in the Nature Portfolio Reporting Summary linked to this article.

### Data availability
The atomic coordinates and the corresponding cryo-EM maps have been deposited in the Protein Data Bank (PDB) and in the Electron Microscopy Data Bank (EMDB) under accession codes 9QSX and EMD-53343 for AQP3 at pH 8.0; 9QSY and EMD-53344 for AQP3 at pH 5.5; and 9QSZ and EMD-53345 for AQP3-H$_2$O. Simulation setup files are available at: [https://github.com/deGrootLab/aquaporin_3_2025]. Source data are provided with this article. Source data are provided with this paper.

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

## Acknowledgements

The cryo-EM data were collected at the Cryo-EM Swedish National Facility funded by the Knut and Alice Wallenberg, Family Erling Persson, and Kempe Foundations, SciLifeLab, Stockholm University, and Umeå University. We would like to thank Julian Conrad, Karin Wallden, and Marta Carroni in SciLifeLab in Stockholm University for cryo-EM grid preparation and data collection, and Anu Tyagi for training in sample preparation. We acknowledge the National Academic Infrastructure for Supercomputing in Sweden (NAISS), partially funded by the Swedish Research Council through grant agreement no. 2022-06725, for awarding this project access to the LUMI supercomputer, owned by the EuroHPC Joint Undertaking and hosted by CSC (Finland) and the LUMI consortium. This study was supported by the Swedish Research Council (2024-02673 to K.L-P. and 2022-06230 to P.H), Diabetesfonden (DIA2020-500 to K.L-P.), the Swedish Cancer Society (23 2746 Pj and 20 0747 PjF to K.L-P.), and the National Sciences and Engineering Research Council of Canada (NSERC) (PGSD-599306-205 to C.J.W.).

## Author contributions

P.H., R.V., C.J.W., I.A., B.dG. and K.L-P. designed the experiments; P.H., R.V., S.B. and I.A. conducted the experiments; C.J.W. performed the simulations; P.H., R.V., C.J.W., S.B, R.P., P.G., I.A., B.dG. and K.L-P. analyzed the data, P.H., R.V., C.J.W., I.A. and K.L-P. wrote the manuscript, with contributions from all authors.

## Funding

## Competing interests

The authors declare no competing interests.
