## [Transparent Peer Review file · Nature Communications]

Structural insights into AQP3 channel closure upon pH and redox changes reveal an autoregulatory molecular mechanism

Corresponding Author: Professor Karin Lindkvist-Petersson

Version 0:

Reviewer comments:

Reviewer #1

(Remarks to the Author)

In this paper, the authors employ structural studies and detailed molecular dynamics simulations to investigate the role of AQP3. At pH 8.0, the AQP3 pore is open in the absence of H₂O₂ but closed in the presence of H₂O₂. The closure of AQP3 pore is attributed to a conformational change in loop E, which is induced by H₂O₂ accumulation and interaction. Furthermore, at pH 5.5 in the absence of H₂O₂, the AQP3 pore is also closed, exhibiting the same conformation as that seen with H₂O₂ at pH 8.0. This low pH-induced closure is attributed to the protonation of D163. The authors focus on beta-cells, investigating the involvement of AQP3 in samples from diabetic patients, and suggest that AQP3 is an important redox homeostasis regulator for beta-cells.

The authors reveal novel insights into the regulatory mechanisms of the AQP3 channel, and a new model of this channel has been proposed. However, there are several points that need to be clarified in the manuscript.

1. The following sentence in the Introduction is difficult to understand: "Notably, intracellular acidification has been observed in artificial β cells under hyperglycemic conditions⁹, suggesting that AQP3 may regulate the transmembrane flux of NOX generated H₂O₂ in human pancreatic β -cells." (lines 51-53). The relationship between low pH conditions and redox homeostasis in beta cells is not clearly established in the Introduction. The authors should provide more information on whether or not such a relationship exists in beta-cells. Intracellular acidification may not be related to AQP3. Although AQP3 is reported to be closed at low pH, this may refer to extracellular low pH.
2. Given the manuscript's Results focus on both low pH and redox effects, the title and Abstract, which currently emphasize redox, might be better revised to reflect the findings related to low pH. This point is related to the comment 1.
3. Glycerol molecules are modeled in the open AQP3 structure at pH 8.0. However, the cryo-EM density corresponding to the glycerols in Fig. S2 is ambiguous. Do the glycerol models fit convincingly into the density maps? The resolution of the map may not be sufficient to model the glycerols unambiguously. The authors should also consider the possibility that some molecules other than glycerols are present in the pores of AQP3.
4. The resolutions of the cryo-EM maps corresponding to fluorinated-foscholine-8 in Fig. S2 and H₂O₂ in Fig. S7 may not be sufficient to unambiguously model these molecules. Is the corresponding density for foscholine-8 observed in the two cryo-EM maps of closed AQP3 structures? Are the modeled positions of the 4 H₂O₂ molecules consistent with the H₂O₂ distribution probability observed in MD simulations?
5. Why does D163 have such a high pK_a value in an open conformation?
6. Do the authors think that the H₂O₂-induced conformational change is initiated by a change in interactions around the residue D163?
7. In Figure 4b, the p-value is 0.16. This suggests that the observed difference in AQP3 expression between normoglycemic and type 2 diabetic donors may not be statistically significant. The sentence "...AQP3 expression was elevated in type 2

diabetic islets...” should be re-examined.

8. The authors need to provide more quantitative analyses for the comparison of AQP3 expression levels by immunostaining in Fig. 4d.

9. The authors should consider experimentally evaluating the effect of the rare R223C mutation of AQP3. Does the rare mutation induce loss of function of AQP3 and cause beta-cell death, leading to the disease? Does the rare mutation change the expression level of AQP3?

10. Please ensure consistent formatting for pH values throughout the manuscript (e.g., “pH 8.0” instead of “pH 8”).

11. In Fig. S1, the legend indicates three models from 3D classification, but only two appear to be shown. Please correct this.

12. Could the addition of 150 μ M H₂O₂ potentially lower the pH of the solution?

13. In Fig. S8, SDS-PAGE of AQP3 does not exhibit MSP. Please clarify whether or not the SDS-PAGE sample is AQP3 reconstituted in nanodisc. Does the size exclusion chromatography profile exhibit AQP3 reconstituted in nanodisc?

Reviewer #2

(Remarks to the Author)

Reviewer #3

(Remarks to the Author)

In this study, Huang et al. combine cryo-EM data and molecular dynamics (MD) simulations to demonstrate pH- and H₂O₂-dependent gating of AQP3. They then apply immunohistochemistry to explore the role of AQP3 in human metabolism.

The work is well executed and clearly presented. The use of nanodiscs for cryo-EM is appropriate and consistent with previous studies showing improved side-chain resolution for AQP and other membrane proteins. The modeling is robust and supports the main conclusions. The immunohistochemistry also appears well conducted, although I am not an expert in that area.

Overall the study is sound and of high impact, and well suited to Nat Comms. I suggest a few additions/clarifications to the text to help guide the reader through the data.

- Were any lipids visible in the EM density maps? This has been an interesting feature in other nanodisc-based EM studies, including for AQPs. A brief mention would be helpful.
- How were the free energy profiles in 1c and e, etc made? Presumably not using the non-equilibrium FEP method described in the Methods. Were they derived from density conversions? Please clarify and include details in the Methods section.
- For the data in Fig 2f, this would benefit from convergence analysis (as has previously been done by this lab).
- How does the Y212 position seen in this study compare to the recent rat AQP3 structure? (<https://doi.org/10.1038/s41467-025-57728-3>)
- Line 91: How were the “rate-based permeabilities” computed? Please elaborate.
- Line 216: The statement that “AQP3 adopted a conformation nearly identical to the acidic (closed) structure” would benefit from a quantitative comparison, such as an RMSD value
- Is fig 3D from MD or EM? “were modelled” could refer to either. Either way, I find it a little hard to compare the MD and EM data here – a direct comparison of the densities from both methods might be clearer
- For the H₂O₂ clusters in Fig 3C from MD, was the same analysis performed for H₂O? If H₂O is also seen in these positions, is there a chance that the EM densities are for this?
- Line 264: does this p value of 0.16 not suggest that there isn’t a significant difference in AQP3 expression in T2D islets?

Version 1:

Reviewer comments:

Reviewer #1

(Remarks to the Author)

Thank you for the revision and responses. The revised manuscript and the associated cryo-EM maps and models have been reviewed.

Major points for correction:

The authors have modeled several small molecules (glycerol, FF8, H₂O₂, and water) in the structures. However, the justification for their placement relies on indirect evidence, such as molecular dynamics simulations and analogies to previously determined aquaglyceroporin structures. This indirect evidence is insufficient to support the modeling of these small molecules, particularly when their binding sites are not well-defined. Greater consideration should be given to the direct evidence from the cryo-EM map itself. Reiterating the previous concern, the resolution of the current cryo-EM map is insufficient to reliably support the placement of these small molecules. The authors should remove the models of these small molecules from the PDB coordinate files. Consequently, the main text and figures related to the modeling of these small molecules must also be removed and corrected.

Minor points for correction:

A hydrogen bond between Ser49 and His53 is described in the results section, but this interaction is not depicted in any of the figures. To aid the reader's understanding, please clearly label Ser49 and depict this hydrogen bond in Figure such as Fig. 2a.

It might be helpful to perform a thorough proofread of the entire manuscript. The following are some examples of apparent errors: The water permeation rate value is inconsistent between the results section (0.67 ± 0.10) and Fig. 1f (0.69 ± 0.09). In the lower panel of Supplementary Fig. S11, H53 (open) should likely be H53 (closed).

Reviewer #2

(Remarks to the Author)

Reviewer #3

(Remarks to the Author)

The authors have fully addressed my comments and I feel that the manuscript is now ready for publication.

Reviewer 1

The authors reveal novel insights into the regulatory mechanisms of the AQP3 channel, and a new model of this channel has been proposed.

We thank the reviewer for their generous summary

1. *The following sentence in the Introduction is difficult to understand: “Notably, intracellular acidification has been observed in artificial β cells under hyperglycemic conditions, suggesting that AQP3 may regulate the transmembrane flux of NOX generated H₂O₂ in human pancreatic β -cells.” (lines 51-53). The relationship between low pH conditions and redox homeostasis in beta cells is not clearly established in the Introduction. The authors should provide more information on whether or not such a relationship exists in beta-cells. Intracellular acidification may not be related to AQP3. Although AQP3 is reported to be closed at low pH, this may refer to extracellular low pH.*

Thank you for this comment, we have now provided more information and the paragraph is updated reading:

“Unlike AQP7, AQP3 facilitates the transport of H₂O₂ and is gated by pH, fully active at neutral pH and completely in-active at pH 6 or lower⁸. Notably, intracellular acidification has been observed in artificial β -cells under hyperglycaemic conditions⁹ in addition to the more well-known acidification of the interstitial fluid in diabetic patients¹⁰, suggesting that the activity of AQP3 may change upon glycemic alterations and possibly effect the transmembrane flux of NOX generated H₂O₂ in human pancreatic β -cells. In addition, glucose in chronic excess causes intracellular oxidative stress, possibly connected to AQP3 activity, which in turn results in defective insulin gene expression and insulin secretion as well as increased apoptosis in pancreatic β -cells¹¹. “

2. *Given the manuscript’s Results focus on both low pH and redox effects, the title and Abstract, which currently emphasize redox, might be better revised to reflect the findings related to low pH. This point is related to the comment 1.*

The title has been changed and now reads:

“Structural insights into AQP3 channel closure upon pH and redox changes reveal an autoregulatory molecular mechanism.”

3. *Glycerol molecules are modeled in the open AQP3 structure at pH 8.0. However, the cryo-EM density corresponding to the glycerols in Fig. S2 is ambiguous. Do the glycerol models fit convincingly into the density maps? The resolution of the map may not be sufficient to*

model the glycerols unambiguously. The authors should also consider the possibility that some molecules other than glycerols are present in the pores of AQP3.

We agree with the reviewer that it is not possible to know exact conformations of the glycerol molecules from the cryo-EM densities found in the channels. However, we are quite confident that the densities represent glycerol molecules, as this is consistent with previously determined structures of aquaglyceroporins, including cryo-EM structure of AQP7 (Huang et al., 2023) as well as high resolution X-ray structure of AQP7 (de Mare et al., 2020). In addition, glycerol was used up to 20% during protein purification thus we do not expect any other molecule than glycerol and water to be present within the channels. Therefore, we have modelled glycerol molecules based on the density and on the knowledge of expected hydrogen bonds where possible. To make this clear to the reader we have added a sentence reading:

“Cryo-EM densities, consistent with glycerol were found in the channels and glycerol molecules were modelled in all four channels (Fig. 1a, Fig. S2). Although we cannot resolve the exact conformations and positions of the glycerols within the channel, we have modelled the glycerol molecules based on the expected hydrogen bond interactions and previous knowledge of aquaporin-glycerol structures. “

4. *The resolutions of the cryo-EM maps corresponding to fluorinated-foscholine-8 in Fig. S2 and H2O2 in Fig. S7 may not be sufficient to unambiguously model these molecules. Is the corresponding density for foscholine-8 observed in the two cryo-EM maps of closed AQP3 structures? Are the modeled positions of the 4 H2O2 molecules consistent with the H2O2 distribution probability observed in MD simulations?*

In the AQP3 pH 8.0 structure we observe an extended density within the central pore. While it is correct that we cannot be entirely sure that this represents FF8, based on the buffer components this is the most likely ligand and therefore we chose to model it within the central pore. We cannot model anything within the central pore in the H2O2 or low pH structures as C4 symmetry was imposed on these models introducing bias within the central pore. However, we have also processed those datasets using C1 symmetry and in those structures, we observe smaller cryo-EM densities within the central pore, where FF8 cannot be modelled.

Regarding the H2O2 positions – we observe clear densities in the closed H2O2 structure, while these densities are not seen in the closed low pH structure. However, it is not possible to conclude that these densities represent H2O2 merely from cryo-EM maps and therefore we have conducted MD analysis, which supported the binding sites for H2O2 in these positions. To make this clearer, we now include an overlay of the cryo-EM densities for H2O2 and calculated H2O2 positions from the MD simulations (Fig. 3).

5. *Why does D163 have such a high pKa value in an open conformation?*

Only two aspartates have pKa values that are significantly shifted from reference (i.e., 3.9): D163 and D219. D163 is buried (SASA ~ 0.29 nm²) and lacks significant charge stabilizing interactions, forming predominantly the hydrogen bond interaction with N209. In the closed state, the SASA increases marginally; however, the stabilizing interaction with N209 is lost which pushes the free energy of deprotonation even higher. D219 is completely buried (SASA ~ 0 nm²), but forms a permanent salt bridge with R223 and backbone and sidechain interactions with T149 that completely stabilize the negative charge. D180 is also buried (SASA ~0.19 nm²), however it forms a salt bridge with R95 and interactions with Y219, resulting in a pKa close to reference.

We have included a figure in the supplemental materials showing the SASA and pKas and updated the text to mention the basis for the D163 pKa shift:

“The underlying basis for this destabilization is that Asp163 is buried, with a low solvent-accessible surface area (Fig. S7), and lacks nearby charge-charge stabilizing interactions, instead forming a hydrogen bond with Asn209 (Fig. 2a). This environment raises the free energy of deprotonation, shifting the apparent pKa in the open state to 7.8 ± 0.2 , substantially elevated relative to the intrinsic aspartate pKa of 3.9 and placing it within the physiological pH range (Fig. 2d). In the closed state, solvent accessibility increases slightly (Fig. S7), but the loss of the stabilizing Asn209 interaction further increases the deprotonation free energy.”

6. *Do the authors think that the H2O2-induced conformational change is initiated by a change in interactions around the residue D163?*

Based on our data we believe that the AQP3 exists in two conformations and can sample between the open and closed states. The acidification can induce the shift towards the closed state due to disruptions of interactions shown by MD simulations. However, we do not think that hydrogen peroxide would necessarily be an inducing factor itself for the changes in the interactions, but would shift the equilibrium towards the closed state by binding to the loop and stabilize the closed conformation of AQP3.

7. *In Figure 4b, the p-value is 0.16. This suggests that the observed difference in AQP3 expression between normoglycemic and type 2 diabetic donors may not be statistically significant. The sentence "...AQP3 expression was elevated in type 2 diabetic islets..." should be re-examined.*

8.

Yes, we are aware that the data does not show statistically significant differences, and that is why we do not use the word "significant". To be even clearer we have re-phrased the sentence, now reading:

"A comparison of AQP3 mRNA levels from normoglycemic and type 2 diabetic donor islets in a larger cohort (n=188) showed that AQP3 expression was elevated in type 2 diabetic islets (Fig. 4b, p value =0.16, the lack of significance is likely due to the small sample size of T2D donors)"

9. *The authors need to provide more quantitative analyses for the comparison of AQP3 expression levels by immunostaining in Fig. 4d.*

We have quantified AQP3 expression levels in islets from normoglycemic and T2D donors. The data is shown in Figure 4e and a paragraph has been added reading:

"Upon quantification of AQP3 expression levels in islets from normoglycemic (n=3) and T2D (n=4) donors, a slight non-significant increase in AQP3 protein expression was observed (p=0.18) (Fig. 4e). Quantification of immunofluorescence revealed a large variation of signal intensity between different T2D donors (p<0.0001). Unfortunately, we could not associate the variation in AQP3 expression to a specific trait due to the low number of T2D donors available and large variation in donor characteristics (like age, BMI, disease onset etc)."

10. *The authors should consider experimentally evaluating the effect of the rare R223C mutation of AQP3. Does the rare mutation induce loss of function of AQP3 and cause beta-cell death, leading to the disease? Does the rare mutation change the expression level of AQP3?*

Thank you for this comment, we agree that it would be valuable to analyse the rare R223C mutation further, but it would require generation and analyses of animal models, thus we believe that it is out of the scope of the current manuscript.

11. Please ensure consistent formatting for pH values throughout the manuscript (e.g., “pH 8.0” instead of “pH 8”).

This has been changed according to suggestion.

12. In Fig. S1, the legend indicates three models from 3D classification, but only two appear to be shown. Please correct this

Thank you for noticing this, we have corrected the mistake.

13. Could the addition of 150 μ M H₂O₂ potentially lower the pH of the solution?

To be certain that the concentration of H₂O₂ used in the experiment does not affect the pH of the sample, we have tested to supplement the final size exclusion buffer with 150 μ M H₂O₂ and this did not change the pH.

14. In Fig. S8, SDS-PAGE of AQP3 does not exhibit MSP. Please clarify whether or not the SDS-PAGE sample is AQP3 reconstituted in nanodisc. Does the size exclusion chromatography profile exhibit AQP3 reconstituted in nanodisc?

Thanks for your questions. Scaffold protein MSP1E3D1 has similar size as AQP3 monomer, approximately 30 kDa, so they might not be differentiated on the gel. We do also compare SEC profiles for AQP3 sample in DDM and nanodisc and a minor shift of the peak can be observed; The sample was run through buffer volume of 24 mL in SEC containing only 20 mM Tris and 100 mM NaCl without any detergents, as observed in Fig. S10, the sample was well behaved in SEC, suggesting AQP3 was reconstituted into nanodisc.

Reviewer 3

Overall the study is sound and of high impact, and well suited to Nat Comms. I suggest a few additions/clarifications to the text to help guide the reader through the data.

We thank the reviewer for their positive feedback

1. *Were any lipids visible in the EM density maps? This has been an interesting feature in other nanodisc-based EM studies, including for AQPs. A brief mention would be helpful.*

Indeed, we observed some extra cryo-EM densities surrounding the transmembrane parts of the protein. We have included a more detailed description of the cryo-EM densities in the method section, where we mention the likely presence of lipids:

“The cryo-EM maps for all the structures allowed to model most of the amino acids, except for the termini. For the pH 8.0 structure, cryo-EM densities were assigned to glycerol and FF8 detergent and in the closed AQP3-H₂O₂ structure four H₂O₂ molecules were modelled. In all the structures additional densities surrounding the transmembrane part of the tetramer could be seen, which most likely represent lipid molecules used in nanodisc reconstitution of the protein.”

2. *How were the free energy profiles in 1c and e, etc made? Presumably not using the non-equilibrium FEP method described in the Methods. Were they derived from density conversions? Please clarify and include details in the Methods section.*

All free energy profiles were constructed using umbrella sampling. Mention of this was somehow excluded from the submitted version of the manuscript, we have updated accordingly:

“For the umbrella sampling simulations, initial configurations were generated by either pulling the solute along the channel axis or manually shifting its z-position. Umbrella windows were spaced at 1 Å, with a 2000 kJ/mol/nm² harmonic restraint applied to the solute’s z-coordinate. Lateral restraints were imposed using flat-bottom potentials in the x and y directions: zero within a 10 Å radius from the channel axis and harmonic (k=1000kJ/mol/nm²) beyond. Each window was simulated for 100-200 ns, discarding the first 10 ns as equilibration. The solute in each of the four monomer channels was treated as an independent replicate. Potential of mean force (PMF) profiles were computed from the resulting probability distributions using the weighted histogram analysis method (WHAM).”

3. *For the data in Fig 2f, this would benefit from convergence analysis (as has previously been done by this lab).*

Indeed, we have included a figure in the supplemental materials showing the converged overlap of the work distributions for the three probed pKa values in the two states and included a sentence

in the main text.

“We observed good overlap in the calculated work distributions (Fig. S11).”

4. *How does the Y212 position seen in this study compare to the recent rat AQP3 structure? (<https://doi.org/10.1038/s41467-025-57728-3>)*

A structural alignment of the published rat AQP3 structure (PDB ID: 8Y8O) with the human AQP3 structure at pH 5.5 is shown in the supplementary file (Figure S4). The two structures align very well (RMSD 0.343) and the tyrosines as well as loop E adopt basically identical conformations. We have included a sentence noting this into the manuscript.

“A similar closed conformation of rat AQP3 with Tyr212 blocking the channel pore was recently reported (Fig. S4)¹¹”

5. *Line 91: How were the “rate-based permeabilities” computed? Please elaborate.*

We apologize that mention of this was somehow excluded from the submitted version of the manuscript, we have updated the methods section accordingly:

“Rate-based permeabilities were calculated assuming a Kramer’s style form of the rate and using the potential of mean force barrier height. Specifically,

$$k \approx \frac{D}{\delta^2} e^{-\Delta G^\ddagger/k_B T}$$

where D is the in-pore diffusion assumed to be 3 times slower for glycerol than water⁴⁴, δ is pore length i.e., 5 nm, ΔG^\ddagger is the maximum barrier height, and $k_B T$ is thermal energy.”

We have also modified the main text to read:

“Assuming a Kramer’s style rate, the barriers suggest that glycerol permeates roughly 500 times slower than water, in good agreement with previous experimental measurements¹¹.”

6. *Line 216: The statement that "AQP3 adopted a conformation nearly identical to the acidic (closed) structure" would benefit from a quantitative comparison, such as an RMSD value*

We have included RMSD value in the brackets next to this statement.

7. *Is fig 3D from MD or EM? “were modelled” could refer to either. Either way, I find it a little hard to compare the MD and EM data here – a direct comparison of the densities from both methods might be clearer*

We have updated Figure 3 (panel c) to more clearly show the clusters identified from the unbiased MD simulations and those modelled from the cryo-EM.

8. *For the H₂O₂ clusters in Fig 3C from MD, was the same analysis performed for H₂O? If H₂O is also seen in these positions, is there a chance that the EM densities are for this?*

The amorphous EM density observed near loop E in the closed + H₂O₂ map is distinct from that seen in the closed apo map, suggesting an enrichment of bound molecules. Our hypothesis from these two closed maps/structures was not that there existed a single well defined ligand binding site, but that H₂O₂ may interact preferentially with loop E which would stabilize the closed configuration.

To test this hypothesis, we performed unbiased simulations with H₂O₂ and, having observed local enrichment and several clusters, switched to umbrella sampling simulations to get a quantitative estimate of the binding. The umbrella sampling does suggest preferential interactions relative to bulk near loop E. Moreover, because the umbrella sampling is performed in the presence of water molecules, the stabilization of H₂O₂ in this region must be more favorable than water, supporting the presence of H₂O₂ at these sites. Although we don't claim exclusive occupancy by H₂O₂, the combination of distinct EM densities, preferential PMF stabilization, and unbiased MD clusters that overlap with the cryo-EM, collectively supports the assignment of an H₂O₂ rich ensemble near loop E. We have updated the main text to reflect this:

“Our simulations do not suggest a unique well-defined H₂O₂ binding site; rather, they suggest a binding region near the collapsed loop E, that likely involves multiple peroxide and water molecules. Binding of H₂O₂ within this region, in turn, stabilizes the protein's closed conformation.”

9. *Line 264: does this p value of 0.16 not suggest that there isn't a significant difference in AQP3 expression in T2D islets?*

Yes, we are aware that the data does not show statistically significant differences, and that is why we do not use the word “significant”. To be even clearer we have re-phrased the sentence, now reading:

“A comparison of AQP3 mRNA levels from normoglycemic and type 2 diabetic donor islets in a larger cohort (n=188) showed that AQP3 expression was elevated in type 2 diabetic islets (Fig. 4b, p value =0.16, the lack of significance is likely due to the small sample size of T2D donors)”

Major points for correction (from reviewer):

The authors have modeled several small molecules (glycerol, FF8, H₂O₂, and water) in the structures. However, the justification for their placement relies on indirect evidence, such as molecular dynamics simulations and analogies to previously determined aquaglyceroporin structures. This indirect evidence is insufficient to support the modeling of these small molecules, particularly when their binding sites are not well-defined. Greater consideration should be given to the direct evidence from the cryo-EM map itself. Reiterating the previous concern, the resolution of the current cryo-EM map is insufficient to reliably support the placement of these small molecules. The authors should remove the models of these small molecules from the PDB coordinate files. Consequently, the main text and figures related to the modeling of these small molecules must also be removed and corrected.

Response:

We appreciate reviewer's concern regarding the placement of small molecules in cryo-EM maps, and we completely support a conservative evaluation of maps and ligand modelling. Indeed, we believe we have modelled the ligands using this type of conservative approach throughout the manuscript. Maybe we were not entirely clear in the text, but the ligand positions are based on the direct evidence, that is cryo-EM densities, which can be clearly seen. To make the interpretation even more transparent we have now included a table with correlation coefficients (CC) calculated by Real-space refine in phenix for each ligand. When compared to the overall CC, ligand values are in line with literature state of the art.

In the AQP3 pH 8.0 structure, where the individual open channels display densities and we modelled glycerol molecules, the CC for the overall structure is 0.85 and most of the glycerol molecules have a CC of 0.7-0.8, which supports the model. Two glycerol molecules have a CC below 0.7, but based on the density it is reasonable to model them in those positions. The same is true for two water molecules modelled in this structure – we see a clear density consistent with water. We believe that removing the glycerol molecules from the models would be misleading for the reader, as each channel pore presents clear densities that would otherwise remain unaccounted for. If we compare to other AQP3 structures recently published at Nat Comm, 8Y8R contains a detergent molecule with CC of 0.77, while 8Y8V contains a phospholipid molecule with a CC of 0.66 (calculated with Real-space refine in Phenix). Both, CC and the densities should be taken into account when assigning ligands and this is accepted as long as conclusions are drawn carefully.

For the AQP3-H₂O₂ closed structure we see clear densities, where we modelled H₂O₂ molecules. Although we cannot be sure based solely on the densities are representing H₂O₂ and not water, the fact that these densities are not present in the equivalent positions in the closed structure without addition H₂O₂ before data collection, strongly support the placement. In addition, the calculated CC for the H₂O₂ were 0.84, 0.78, 0.65 and 0.76. Hence, this is direct evidence for the presence of H₂O₂. To even further confirm this, we perform MD simulations, and the results show excellent agreement between the cryo-EM maps assigned for H₂O₂ and the major H₂O₂ clusters calculated during the simulations. Again, we believe that removing these molecules from the models would be misleading to the reader, as our conclusion is that hydrogen peroxide binds in these positions in the protein leading to stabilization of the closed conformation.

Lastly, we modelled one water molecule in the AQP3 pH 5.5 structure interacting with a Tyr212 within the channel pore. The CC for this molecule is relatively low – 0.58, however the density is very clear and possibly there should be more than one water molecule. We have also modelled a few water molecules in the AQP- H₂O₂ structure, in positions supported by the cryo-EM density within the channel. We include additional clarifications about our modelling of water in the method section.

We think that our modelling approach is accepted within the cryo-EM field and is conservative when placing the ligands. Placement of water molecules has been reported based both on cryo-EM map and MD calculations previously (Hun-Roh et al., 2020, Science Advances, pdb id 6M0R). In this study, water pathway was modelled with CC for water molecules ranging from 0.2 to 0.8 and for the most of them in the range of 0.5-0.6 (calculated with Real-space refine in Phenix). Another recent study in

Nature Comm (Kashyap et al., 2025) report a di-copper binding site, where the CC for copper ions is around 0.5 and for adjacent water molecules as low as 0.2-0.3 (calculated with Real-space refine in Phenix), illustrating once again that we followed a more restrictive approach for modelling and that in many cases the modelling should be based not only on the direct evidence from cryo-EM maps but also supporting data to be able to draw correct conclusions. Accordingly, all ligand positions presented are strongly supported by the cryo-EM data and therefore we would prefer to keep these molecules in the PDB files.

Minor points for correction:

A hydrogen bond between Ser49 and His53 is described in the results section, but this interaction is not depicted in any of the figures. To aid the reader's understanding, please clearly label Ser49 and depict this hydrogen bond in Figure such as Fig. 2a

We thank the reviewer for this comment and Ser49 is now shown as sticks in figure 2.

It might be helpful to perform a thorough proofread of the entire manuscript. The following are some examples of apparent errors: The water permeation rate value is inconsistent between the results section (0.67 ± 0.10) and Fig. 1f (0.69 ± 0.09). In the lower panel of Supplementary Fig. S11, H53 (open) should likely be H53 (closed).

The spotted mistakes have been changed, and the manuscript has been proofread.